# Autism with Epilepsy: A Neuropsychopharmacology Update

**DOI:** 10.3390/genes13101821

**Published:** 2022-10-08

**Authors:** Roberto Canitano, Roberto Palumbi, Valeria Scandurra

**Affiliations:** 1Division of Child and Adolescent Neuropsychiatry, University Hospital of Siena, 53100 Siena, Italy; 2Basic Medical Sciences, Neurosciences, and Sensory Organs Department, University of Bari “Aldo Moro”, 70124 Bari, Italy

**Keywords:** autism, epilepsy, treatment, child and adolescent

## Abstract

The association between autism spectrum disorders (ASD) and epilepsy has been extensively documented, and the estimated prevalence varies depending upon the selected population and the clinical characteristics. Currently, there are a lack of studies assessing the patient care pathways in ASD, particularly for comorbidity with epilepsy, despite its personal, familial, and economic impacts. Genetic abnormalities are likely implicated in the association of ASD and epilepsy, although they are currently detectable in only a small percentage of patients, and some known genetic and medical conditions are associated with ASD and epilepsy. There is no specificity of seizure type to be expected in children and adolescents with ASD compared with other neurodevelopmental disorders or epileptic syndromes. Treatment options include antiepileptic drugs (AED) and developmentally-based early interventions for ASD. Carbamazepine and lamotrigine are the most used AED, but further studies are needed to more precisely define the most suitable medications for this specific group of children with ASD.

## 1. Introduction

Autism spectrum disorder (ASD) and epilepsy are two conditions characterized by a high rate of comorbidity, sharing several common risk factors [1,2,3]. Recent data from the Centers for Disease Control and Prevention (CDC) and the Autism and Developmental Disabilities Monitoring (ADDM) Network identify the prevalence of ASD as 1.5–1.7%, thus, remarking that it is a common disorder in childhood [4]. The reported prevalence of epilepsy in autism was much lower using former constricted criteria, than its prevalence defined by the updated version of the *Diagnostic and Statistical Manual of Mental Disorders* (DSM 5) [5]. If children with intellectual disabilities (ID), frequently occurring in autism (intelligence quotient [IQ] < 70), are excluded, the prevalence is still expected to be greater than that in the general population [5].

Significant differences in the rates of epilepsy in autism have been reported, especially in relation to cognitive levels, depending on the different study methods employed, including the study population and definitions of the disorders. In a recent study, a prevalence of comorbidity between ASD and epilepsy was reported in 21.5% of patients. Moreover, the rates of comorbidity were higher in ASD patients with intellectual disability (ID) [3]. Similar data were also reported by another meta-analysis that compared the prevalence of the co-occurrence of ASD and epilepsy, considering the presence of ID and the age of the patients (under or over the age of 12) [1]. A recent systematic review and a meta-analysis reported the updated prevalence data about the comorbidity between epilepsy and ASD (from 9% to 19% of ASD patients) [2]. Moreover, these studies described an increased prevalence of epilepsy in adolescents with autism (11–17 years old), and a rising rate of comorbidity among pre-school children with autism. The prevalence of epilepsy increased with age, in females, and in autistic individuals with low intellectual function. New updated figures on the prevalence of epilepsy in autism have, thus, been provided [2].

Considering the influence of epilepsy on the core symptoms of autism, a study using the Social Responsiveness Scale (SRS) found that ASD patients with epilepsy showed a significant impairment in social interaction/communication scores. In fact, ASD patients with epilepsy also scored significantly higher on total SRS t-score than ASD patients without epilepsy [6]. An interesting study investigated the independent risk of four ASD severity measures and how they might impact the comorbidity with epilepsy (ID, language impairment, core ASD symptom severity, and motor dysfunction) [7]. It was found that the presence of ID, language abnormalities, ASD-specific symptom severity, and motor issues independently predicted a small increased risk for epilepsy. A small effect size in the association between epilepsy and developmental regression was also found. In conclusion, the ASD severity markers investigated in the risk of epilepsy demonstrated that the association of autism with epilepsy is due to different factors and not only to the effect of ID [7].

Except for the cases where seizures occur in patients affected by epileptic syndrome who later also develop autistic symptoms, the occurrence of seizures in children and adolescents with ASD is quite unpredictable; in addition, seizures represent clinical urgency in ASD patients. Epilepsy must be considered by clinicians and family members as soon as it develops, because some interventions should be made readily available to the child. Even if focal or localization-related seizures are often reported, there is no peculiar association between ASD and seizure type or a specific epileptic syndrome. Seizure might have an early onset during infancy but also in adulthood, with some evidence of a bimodal age distribution. The severity of epilepsy in populations with ASD has not yet been precisely defined, and there has been very little investigation of the role of epilepsy on autism core social impairment. Moreover, the risk of autism in patients with epilepsy has been less studied, but there is evidence that ASD risk is greater in those with epilepsy, when compared with the general population [8,9,10].

## 2. Genetics of ASD with Epilepsy

According to a current hypothesis, common diseases are the result of multiple rare genetic variants that have great functional effects, and this hypothesis perfectly fit with the comorbidity between ASD and epilepsy. Several genetic abnormalities such as copy number variants (CNVs; e.g., microdeletions, microduplications, insertions) and single gene disorders have been associated with both ASD and seizures, but these alterations are still under evaluation [10]. In addition, more than 100 disease genes have been described as related to ASD, including SHANK3, CNTNAP2 and NLGN4X, and many of them are also implicated in epilepsy [11].

Genetic abnormalities involved in the risk of autism with epilepsy are within the overall number of genes relevant to ASD, as described in recent work [11,12,13] (see Table 1). The current hypothesis is that epilepsy and ASD might share common gene risk factors that could finally interfere with normal neurodevelopment [14,15,16,17].

Molecular abnormalities in synaptic structures and functions in ASD and epilepsy frequently involve neuroligins and neurexins, proteins that are crucial for aligning and activating synapses along with the SHANK3 scaffolding protein [18]. Multiple genes may contribute to the disruption of GABAergic interneuron development, which are involved in both autism and epilepsy. Mutations in GABA-A receptor subunit genes have been associated with ASD. The genes coding for the three GABA-A receptor subunits α5, β3 and γ3 (GABRA5, GABRB3 and GABRG3, respectively) are located on the 15q11chromosome, and single nucleotide polymorphisms (SNPs) in these genes have been associated with ASD [19]. Neurexins (NRXNs) are presynaptic proteins that bind their postsynaptic counterparts, the neuroligins (NLGNs). NRXN-NLGN signaling is consistently involved in postsynaptic differentiation and it controls the balance of inhibitory GABAergic and excitatory glutamatergic signaling. Mutations and chromosomal rearrangements in NRXN have been associated with ASD [20], and it has been shown that NRXNs can bind not only NLGNs but also GABAA receptors, with the result of decreasing GABAergic transmission. NLGN1, NLGN4X and NLGN4Y neuroligins are localized at glutamatergic synapses, while NLGN2 is located in GABAergic synapses [21]. Mutations in NLGN1, 3 and 4X genes have been identified in ASD, and two X-linked neuroligin mutations have been associated with familial ASD [22]. As a result, these studies point to abnormal synaptic GABAergic signaling, as relevant examples of molecular abnormalities in ASD and epilepsy.

To further analyze the association of autism with epilepsy, multiplex ASD families were studied, providing evidence of the association of the two disorders. In a study investigating the familial association, the prevalence of epilepsy was 12.8% in individuals with ASD, and 2.2% in siblings without ASD. The risk of epilepsy in multiplex autism was significantly associated with ID, but not with gender. In addition, genetic or non-genetic identified risk factors of autism tended to be significantly associated with epilepsy. When children with prematurity, pre- or perinatal insult, or cerebral palsy were excluded, a genetic risk factor was reported for 10.2% of children with epilepsy, and 3.0% of children without epilepsy (*p* = 0.002). Furthermore, the epilepsy phenotype co-segregated within families (*p* < 0.0001). As a result, epilepsy in multiplex autism likely has significant genetic components and it may define a different subgroup of clinical characteristics and genetic risks [13].

## 3. GABA Abnormalities

Alterations of neocortical minicolumns have been demonstrated in ASD (e.g., morphological aberrations of GABA interneurons). GABAergic interneurons were shown to surround the axons and dendrite of pyramidal neurons among the minicolumns array. A significant decrease in GABA receptors was also detected, further emphasizing the abnormal GABAergic signaling in autism with epilepsy [21,22,23,24].

## 4. Social Difficulties

Social cognition is comprised of a wide range of interrelated functions that include gaze and face processing, emotion recognition, visual fixation to social stimuli and detection of biological motion [25,26]. Nevertheless, more complex social cognitive processes should be considered including social orientation and judgments, perception of social cues and attribution of mental states (e.g., theory of mind (ToM)). Although ToM is generally considered to be an independent cognitive domain, it has connections with overall cognition. Amygdala, insular cortex, and basal ganglia are involved in the process of emotions processing as a developmental process increasing over time [27,28,29,30]. Social cognition is broadly defined as the ability to process social–emotional information and apply it to social situations. Processing social information is based on the coordinated function of a wide neural network mentioned above which has been referred to as the social brain [30]. In ASD, the social deficits have been linked to the amygdala, the superior temporal sulcus, and the fusiform gyrus [14]. Social cognition is primarily affected in ASD, based on abnormal connectivity within the social brain areas, and calls for early intervention to correct the developmental trajectory [25,26,31]. Another defining feature commonly associated with social-cognitive deficits in children with ASD is *joint attention*, defined as the group of complex behaviors used to share the experience of objects or events with others. The development of joint attention is closely related to the development of ToM, which is to understand our own thoughts, intentions, beliefs, and emotions, and to recognize that others have beliefs, intentions, and perspectives that are different from one’s own [27,30]. 

In a recent study, researchers performed a comparative analysis of social skills, such as emotion recognition (ER) and theory of mind (ToM), in children and adolescents affected by epilepsy, ASD and specific learning disorder (SLD). Interestingly, they found that all three clinical groups were significantly impaired in ER and ToM skills when compared with the control group [29]. Social cognitive deficits independent of nonsocial cognition is an important variable affecting developmental outcomes in children with epilepsy and ASD. The multiple genes and molecular pathways shared by epilepsy and ASD indicate a common origin that explains the co-occurrence of both disorders, in addition to ID. Furthermore, social cognition and epilepsy share not only molecular pathways, but also overlapping anatomical circuitry [16,18]. The relationship between epilepsy, ASD and social cognition, has a bidirectional framework that has to be carefully investigated as it still has many unresolved questions [27,28]. 

In conclusion, all these findings support the evidence that the early identification of social deficits in epileptic patients is crucial for better management and care of patients affected by epilepsy and neurodevelopment disorders such as ASD.

## 5. Electroencephalography and Epileptiform Abnormalities

Electroencephalography (EEG) is a primary means for investigating this comorbidity; it is relatively easy to perform and to access. It was administered on a regular basis in this study. EEG in autism is part of the investigation during the diagnostic phase, although it is not specified in the current practice parameters of the AACAP. There are also discordant opinions about the frequency and time of administration at follow-up [30]. However, as cited in some studies, the significant comorbidity between epilepsy and ASD strongly suggests EEG use in case of suspected seizure [32,33]. Further EEG research in ASD has yielded promising results in early detection and prediction of atypical brain development using quantitative techniques and in longitudinal studies to define the developmental stages of the disorder [32,33,34,35,36,37]. Epileptiform abnormalities occur as frequently as 20–30% in individuals with ASD and epilepsy, and should be carefully sought. EEG administration would meet the need for evidence-based precision medicine for all children with autism and epilepsy, providing an instrumental, easy to access evaluation tool [38]. 

Even if paroxysmal epileptiform abnormalities might be involved in cognitive deficits and in the core autistic symptoms, this relationship needs more investigation. Moreover, a higher presence of epileptiform activity was found in ASD children with more severe stereotypes and/or aggressive behavior; on the other hand, the incidence of epileptiform abnormalities was significantly lower in higher functioning ASD individuals [39,40,41]. Further use of EEG to assess a possible ASD biomarker must consider individual differences, including sex; females with ASD and epilepsy are the main example of sex differences, as they are more strongly affected than males [42]. Autism with regression is a condition presenting in about 30% of children and can be related to epileptic syndromes and/or seizure disorders. This has been detailed and must be kept in mind when considering autism with epilepsy [43,44].

## 6. Treatment and Interventions

Early treatments on social deficits should be started quickly after diagnosis for children with a possible association of ASD and epilepsy, because it has been observed that early interventions can consistently modify the outcome [45]. For example, one well-known intervention is the Early Start Denver Model, a psychoeducational treatment that is effective for the improvement of autistic symptoms and global functioning in patients who receive early care [28]. However, the patients of this study were children affected only by ASD, without epilepsy [28].

## 7. Current Appraisal of ASD with Epilepsy

The association of autism with epilepsy has been found in a significant proportion of children and this suggests the need for early detection and management of this comorbidity to improve treatment and outcomes [1,2,3,4]. Young patients already facing all the difficulties related to the primary social and communication disturbances of ASD are exposed to the additional clinical difficulties deriving from seizures [28]. Trends for sex were detected with a 3:1 (76.8% vs. 23.6%) of males/female ratio of children with autism and epilepsy, which is in line with the well-recognized higher prevalence of autism in males. However, according to other reports, this trend might be inverted in other samples of ASD patients with epilepsy [28,46,47]. Two age peaks have been reported for seizures to appear in autism—infancy and adolescence—but with the chance to appear at other points in this age span [43]. 

Epilepsy is a further problem and burden in ASD management [46]. The outcomes of epilepsy in ASD patients is very unpredictable, from benign forms to severe drug-resistant forms [42]. An accurate EEG evaluation in these patients is crucial for a better outcome for both ASD and epilepsy [42,43]. 

## 8. Antiepileptic drugs in ASD

The use of antiepileptic drugs (AED) must follow the clinical features and current guidelines for epilepsy [47,48,49,50]. According to the most recent ILAE updated guidelines, Table 2 summarizes the indications for the initial treatment in monotherapy of the seizures [48,49]. It must be specified that these treatment choices are not autism-specific, and they can be applied for any patient affected by epilepsy.

Currently, there have been no controlled clinical trials which evaluate the effectiveness or efficacy of antiepileptic drugs for the treatment of seizures in subjects with ASD, despite the strong association of these conditions [50]. Two survey studies, one controlled and one uncontrolled, evaluated treatments used for seizures in individuals with ASD. According to the first study, 15.2% of ASD patients received AED treatments, including carbamazepine, valproic acid, and phenytoin, with a high parent-reported satisfaction of the treatment [49]. Frye et al. conducted a retrospective national case-control survey study to assess if the use of several treatments, including AEDs, was more effective than other treatments for patients with ASD and seizures and/or epilepsy. A total of 733 parents of children with ASD who had seizures, epilepsy, and/or an abnormal EEGs were included, along with 290 parents of children with ASD without EEG abnormalities, as a control group. In the sample of children with ASD and clinical seizures, AED treatments were perceived as more beneficial for the seizures but not for other clinical factors. Moreover, four AEDs (valproate, lamotrigine, levetiracetam, and ethosuximide) were reported as effective treatment for the seizure control but they also appeared to worsen other clinical factors [51]. No significant improvement in seizure control was reported for the other AEDs examined (phenytoin, clonazepam, carbamazepine, oxcarbazepine, topiramate, gabapentin, zonisamide, felbamate, and phenobarbital); phenobarbital was characterized by the worst effect on attention, behavior, mood, and sleep. Further in a case series of 66 children with epilepsy, 5 with ASD, and 16 with ASD-like symptoms, valproate, carbamazepine, and ethosuximide were reported as beneficial with a low incidence of adverse effects, in contrast with clonazepam, phenytoin, phenobarbital, and nitrazepam, that were found to have a high prevalence of behavioral adverse effects [52]. 

It is worth mentioning again that because of their neurological and non-neurological adverse effects, the use of AEDs might be limited or contraindicated. Unfortunately, these adverse effects appear to be quite prevalent in patients with developmental disabilities, including ASD [51,52,53,54]. In addition, it is well known how long-term treatment with AEDs is associated with other adverse effects such as memory and/or attention impairment, or psychomotor alterations; all these conditions must be strictly monitored in a fragile population such as patients affected by neurodevelopment disabilities and epilepsy. 

A recent U.S. national survey reported the effectiveness of several treatments (psychopharmacological and non-psychopharmacological interventions) in patients affected by ASD [54]. The non-pharmacological treatments included dietary supplements and psycho-educational interventions, while psychotropic treatments included AEDs and other psychiatric medications. The survey found that: (1) clonazepam and valproate were the most commonly used; (2) the most effective impact was obtained by lamotrigine and oxcarbazepine, followed by other AEDs such as diazepam, levetiracetam, valproate, clonazepam, carbamazepine, gabapentin, and topiramate; (3) lamotrigine, levetiracetam, and oxcarbazepine resulted effective not only in seizures control, but also for anxiety, and aggressive behavior. On the other hand, both clonazepam and gabapentin improved anxiety symptoms, and topiramate was effective for aggressive behavior and cognition; however, these AEDs were not significantly effective for seizures control. Lastly, lamotrigine and oxcarbazepine resulted effective for both seizures and behavioral symptoms of ASD [53]. A systematic review of one randomized clinical trial and four open-label studies for topiramate indicated that it significantly improved irritability and hyperactivity and reduced anxiety and depression, but often caused decreased appetite, agitation, hyperactivity, and cognitive difficulties [54]. However, these results were inconsistent across studies and should be accepted with caution. Additional research is needed to obtain further information on the treatment of autism with epilepsy and on the use of AED for ASD core symptoms [55,56] Table 3.

Beyond the primary use as an AED, valproate has been tested in ASD to reduce irritability with scarce or inconsistent results. Valproate trials have demonstrated moderate effectiveness on irritability in ASD [57,58,59,60], but this has not yet been replicated in larger studies, which are urgently needed to confirm these findings. In an open-label trial and in case reports, valproate resulted quite effective for core ASD symptoms, with few adverse effects on cognition [60,61,62,63]. Moreover, valproate has been indicated for seizures control and for behavioral problems, while carbamazepine, clonazepam and lamotrigine have been recommended only for seizures control. In a study by Mendez et al. [64], valproate was administered to 44% (*n* = 12) of patients, in monotherapy or polytherapy, as an antiepileptic. Most of the children in this study were on monotherapy (76%), while 23% were on polytherapy, in Spain, which suggests a different treatment approach is being taken in the country.

In a randomized double-blind study and one open-label study, it was found that levetiracetam was effective for seizures control, hyperactivity, and inattention, but it was associated with aggressive symptoms and other or behavioral abnormalities [65,66]. Belsito et al. [67] developed a randomized, double-blind, placebo-controlled trial, evaluating the potential benefits of lamotrigine on autistic symptoms in a sample of ASD patients. At the end of the trial, they did not find any statistically significant improvement of the behavioral and socio-communication symptoms between placebo and lamotrigine groups. 

Lastly, oxcarbazepine and gabapentin are recommended for seizures as a second or third line treatment; however, there is still no evidence of their effectiveness for behavioral symptoms [48,68].

## 9. Conclusions

The relevant data obtained by the analyzed studies emphasizes the urgent need to start specific clinical trials on the treatment of autism with epilepsy. Survey data warrant follow-up or replication for confirmation in adequate, powered, scientific studies in order to obtain a clearer picture of recommended treatments. Current approach relies mostly on ILAE and NICE guidelines for epilepsy in childhood, given the absence of clinical guidelines specific to the assessment and management of epilepsy associated with autism. The absence of empirical data to rely upon, and limited data about the efficacy and tolerability of AEDs in ASD, leave a gap that needs to be filled [61]. In many cases, AED choice was not substantiated by seizure characteristics and epilepsy syndrome definition, and only partial information was gathered, especially on seizure types. All this information is required to choose the most suitable treatment for every single child with autism with epilepsy, and to start the development of clinical guidelines. These would address personalized treatment plans that may help caregivers, families and patients to avoid delays or receiving inappropriate interventions. A precision medicine approach following advances in the understanding of specific conditions and the genetic risk of the co-occurrence of autism with epilepsy is now readily expected. The heterogeneous autism population identified so far, challenges the current treatment recommendations. In addition, it is quite complex to include patients affected by ASD and epilepsy into clinical trials. Therefore, the development of new pharmacological treatments and the focus of clinical trials should be informed by pragmatic methodologies to make feasibility the primary objective of such studies, before moving forward to specific treatment choice evaluation.

## Figures and Tables

**Table 1 genes-13-01821-t001:** Examples of genes linked with both ASD and epilepsy. (Modified from Amiet et al., 2013, and Betancur, 2011).

Gene	Locus	Type of Mutation	Transmission	Molecular Abnormalities
*SCN1A*	2q24	Point mutation	De novo	Na^+^-channel
*SCN2A*	2q23–q24.3	Deletion	De novo	Na^+^-channel
*KCNMA1*	10q22	Point mutation	Dominant inheritance	K^+^-channel
*NLGN4X*	Xp22.31	Point mutation	Inherited	Synapse formation
*NRXN1*	2p16.3	Deletion Point mutation	Recessive inheritance De novo	Synapse formation
*CNTNAP2*	7q35	Deletion Point mutation	Recessive inheritance De novo	Synapse formation
*SYNGAP1*	6p21.3	Point mutation	De novo	Synapse RasGAP
*ARX*	Xp22.13	Duplication	Inherited De novo	Aristaless-related homeobox (ARX) protein

**Table 2 genes-13-01821-t002:** Treatment choice in monotherapy for seizures in ASD.

Treatment	Focal Seizures	Absence Epilepsy
First line	Carbamazepine, lamotrigine, levetiracetam	Ethosuximide, valproate
Second line	Topiramate, oxcarbazepina, valproate	Lamotrigine
Third line	Zonisameide, clobazam	
Other	Phenyotin, phenobarbital (good efficacy/poor tolerability)	

**Table 3 genes-13-01821-t003:** Neuropsychopharmacology studies in ASD and epilepsy.

Study	Number	Age	Diagnosis	Type of Epilepsy	Treatment Used	Response to Treatment
Coleman et al., 2019	505	Child (<13) 54% Teenager (13–18) 21% Young adult (19–30) 17% Adult (>30) 8%	ASD	N/A	Clonazepam, valproate, lamotrigine, oxcarbazepine, carbamazepine, diazepam, topiramate, levetiracetam, gabapentin	Lamotrigine and oxcarbazepine had the highest net benefit followed by diazepam, levetiracetam, valproate, clonazepam, carbamazepine, gabapentin, and finally, topiramate, which had a slightly negative net benefit
Hirota et al., 2014	171	8.54 years	ASD	N/A (epileptiform activity, even without clinical seizures)	Valproate, lamotrigine, levetiracetam, and topiramate	No significant global behavioral improvement
Frye et al., 2011	733	Mean age: 13 years 4 months (seizures group) 9 years 5 months (control group)	Children with ASD and clinical seizures, subclinical epileptiform discharges	Generalized seizures, partial seizures, absence seizures	Cluster 1: valproate, lamotrigine, levetiracetam, ethosuximide. Cluster 2: phenytoin, clonazepam, carbamazepine, oxcarbazepine, topiramate, gabapentin, zonisamide, felbamate.	In cluster 1, improvement in seizures was significantly higher than cluster 2

ASD: autism spectrum disorder; N/A: not applicable.

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
