# Peer review of "Autism with Epilepsy: A Neuropsychopharmacology Update"

_genes, 2022, doi:10.3390/genes13101821_

Round 1
Reviewer 1 Report
This is an original article dealing with the challenge to treat patients with both ASD and epilepsy. Such a work will be of interest for a lot of health professionals involved in neurodevelopment, and of course the scientific community. The article includes an extensive review of the literature, and highlights the main challenges of this area of research. Interestingly, it also evokes some proposals and highlights the need for more data in this field.
Major comments:
Line 9-11: neurodevelopmental disorders with a genetic cause are a major nosologic group and this sentence tends to mean that genetic causes of epilepsy and ASD are very rare. I would suggest to modify this sentence and simply insist on the fact that genetic causes can be responsible for number of ASD and epilepsy cases.
Given the strong genetic focus of The Journal on Genetics, the part on neurogenetics might be reviewed, to include more data and evoke the huge number of patients suffering both disorders.
Table 1: there is thousands of ID genes, part of them are also responsible for epilepsy. Either a more formal analysis of these overlapping groups would be helpful here, or the Table 1 (with some formatting) can be called something like “examples of genes responsible for both ASD and epilepsy”.
Line 80: “CNV have been associated to ASD” can you precise? are you mentioning cohorts or high penetrance, possibly recurrent, disease-causing CNV in a patient with ASD; The part on the genetics of ASD seems uncleaur to me (line 79).
Table 2: maybe some basic formatting (bold characters, lines) would help for a quick understanding for the reader for this table.
Minor comments:
2: title: shall it be a dash there ?
3. Authors affiliations: it would be helpful to have more information, like the Department for example, and a few contact information (address, zip code);
Is there a corresponding author?
19: Abbreviation: maybe include “and epilepsy” would better represent the article ?
122: the Social difficulties part is interesting but very short and would merit a to be further developped.
A separated conclusion section would be interesting
Genes names shall be italicized in the text.
Font and size are not always the same (159)
277-278: the two sentence seem unclear
Finally, would it be interesting to mention the doses (per kg for example) for the anti-epileptic drugs ?
Author Response
line 9-11
genetic causes implication has been more clearly detailed and considered as a major contribution in the comorbidity of ASD and epilepsy. Additional data and examples have been included to make it clearer to the reader see page 3
table 1 has been renamed as suggested, given the high number of genes potentially involved in autism with epilepsy , we provided only a few examples and tried to make it easily read.
The role of CNV in ASD have been detailed and clarified.
table 2 has been formatted as previously presented only in a draft form
the manuscript was deeply revised and integrated with data and comments in all the parts marked by the reviewers, also additional paragraphs were added including the social difficulties section at page 4
By the way including the dosage of the drugs mentioned would be helpful and well accepted but currently it cannot be derived by the scarse references on the topic of ASD and epilepsy. More trials and research are needed to provide reliable guidelines on dosage and administration of medications for this condition.
Many thanks to the reviewer for the comments on the manuscript and for helpful suggestions on critical issues of this topic. We hope that this revised form meets the standard for publication on this Journal

Reviewer 2 Report
The review article is a paper that succinctly assesses the correlation between autism spectrum disorders (ASD) and epilepsy and addresses the AED treatment of such patients. The paper is interesting, but needs some revisions and content enrichment:
1. The literature reference is missing in several places, e.g.
- page 2 line 32
- page 2 line 40
- page 5 chapter: : Current appraisal of ASD with epilepsy
- 1st page 6 lakapit 214-221
2. Page 2 line 40 - needs font change to the one used in the rest of the paper
3. In the chapter: Antiepileptic drugs in ASD studies are described - it would be useful to add a table based on these studies, which would include details e.g. characteristics of the patient group (age, diagnosis, type of epilepsy), how large a group of patients was studied, what treatment was used, how long the treatment lasted, response to treatment, etc.
Author Response
Thank you for your comments and helpful suggestions
the references were checked and updated according also to some changes in the list of references with the aim to integrate and expand it. We hope it is clear and useful for the reader,
As to the new table suggested, table 3, we performed once again a checking of the current literature on ASD with epilepsy and still we found just a few data. There is an overall a lack of information that prevent conclusions and guidelines and call for additional research. We presented only the principal studies available with some basic information.
Once again we appreciated revier's comments and hope to have more data in the upcoming future to answer the related questions.
